# Characteristic Assessment of Angiographies at Different Depths with AS-OCTA: Implication for Functions of Post-Trabeculectomy Filtering Bleb

**DOI:** 10.3390/jcm11061661

**Published:** 2022-03-16

**Authors:** Man Luo, Yingting Zhu, Hui Xiao, Jingjing Huang, Jin Ling, Haishun Huang, Yiqing Li, Yehong Zhuo

**Affiliations:** 1State Key Laboratory of Ophthalmology, Guangzhou 510060, China; luom66@mail2.sysu.edu.cn (M.L.); zhuyt35@mail.sysu.edu.cn (Y.Z.); xiaoh59@mail.sysu.edu.cn (H.X.); hjjing@mail.sysu.edu.cn (J.H.); jinling5@mail.sysu.edu.cn (J.L.); huanghsh9@mail2.sysu.edu.cn (H.H.); 2Zhongshan Ophthalmic Center, Sun Yat-sen University, Guangzhou 510060, China; 3Guangdong Provincial Key Laboratory of Ophthalmology and Visual Science, Guangzhou 510060, China; 4Guangdong Provincial Clinical Research Center for Ocular Diseases, Guangzhou 510060, China

**Keywords:** optical coherence tomography angiography, anterior segment, vascularization, angiographic biomarkers, filtering bleb

## Abstract

This study aimed to analyze the quantitative vascular biomarkers of filtering bleb function at different depths using anterior segment optical coherence tomography angiography (AS-OCTA). This cross-sectional study is registered on Clinicaltrails.gov (NCT 04515017). Forty-six eyes with primary open-angle glaucoma that had undergone trabeculectomy with mitomycin-C for more than six months were included. Vessel density (VD) and vessel diameter index (VDI) in the superficial layer (SL), Tenon’s layer (TL), and deep layer (DL) of the bleb were obtained. The VD and VDI were higher in the failure group (both *p* = 0.000). Significant correlations were found between the SL, TL, DL’s VDI, and IOP in the success group (*p* = 0.013, 0.016, 0.031, respectively). The VD of the TL and DL were related to IOP in the failure group (*p* = 0.012, 0.009). Tenon’s VD (TVD) and Tenon’s VDI (TVDI) correlated with IOP adjusting for TVD, TVDI, and the Indiana Bleb Appearance Grading Scale (IBAGS) (*p* = 0.009, 0.043) or Kenfeld grading system (KGS) (*p* = 0.011, 0.016). The area under curve (AUC) of the TVD, TVDI, IBAGS, and KGS to predict surgery failure were 0.960, 0.925, 0.770, and 0.850. AS-OCTA realized the quantitative evaluation of vessels, especially the invisible vascularity beneath the conjunctiva. TVD and TVDI as detected by AS-OCTA better reflected bleb function than conventional grading systems.

## 1. Introduction

Glaucoma is the world-leading cause of irreversible blindness, and the progressive damage of the optic nerve and corresponding visual field defect constitute the characteristic changes of glaucoma [1,2,3]. If the intraocular pressure (IOP) cannot be controlled by medication or laser treatment, trabeculectomy remains the standard surgical procedure to reduce the IOP [4]. The aqueous humor can be drained to sub-Tenon’s space through filtering bleb after the surgery [5,6]. Scarring of the surgical area is the most common cause of surgical failure. Therefore, the evaluation and maintenance of filtering blebs are crucial for the successful outcome of trabeculectomy.

Wound healing and scar formation are accompanied by angiogenesis [7,8,9]. Maintaining the function of postoperative filtering blebs is associated with the degree of conjunctival fibrosis, tissue remodeling, and angiogenesis in the surgical area [10]. Previous studies have shown that vascular formation in the bleb site, especially between the scleral flap and conjunctiva, is the critical factor of scar formation, stabilizing after six months postoperatively [11]. An objective indicator, which can assess the vascular status of the bleb objectively, could help evaluate the filtering bleb function and predict the surgical outcome [12,13,14].

A variety of classification systems have been established based on slit-lamp microscopy (SLM) to provide important information about the status of the bleb, including the Kenfeld grading system (KGS) and the Indiana Bleb Grading Appearance Scale (IBAGS) [15,16,17,18]. These systems involve the evaluation of the height, range, location, surface vascular distribution, and Seidel test of the bleb. They cannot detect the internal structure of the operative region or provide an objective evaluation with a quantitative parameter of the filtering bleb. Recently, ultrasound biomicroscopy (UBM) [19,20] and anterior segment optical coherence tomography (AS-OCT) [21,22] have been applied to show the internal structure of bleb, such as bleb height, internal cystic structure, and thickness of the scleral flap [23,24,25,26]. However, UBM requires contact inspection, and neither of them can imagine the vessel. In previous clinical studies, in vivo confocal microscopy (IVCM) [13,27,28] and vascular fluorescence angiography (VFA) [22,29] have been used to evaluate the vessels of the filtering bleb. In these studies, a highly vascularized bleb was an indicator of surgical failure. However, neither of these imaging techniques provide a quantitative assessment of blood vessels, and both of them require eye contact (IVCM) or contrast injection (VFA) for the procedure.

An objective and quantitative assessment of vascular features has been achieved by optical coherence tomography angiography (OCTA) noninvasively [30,31,32,33]. Akagi et al. successfully visualized conjunctival and intrascleral blood vessels in normal eyes using anterior segment OCTA (AS-OCTA) [34]. Yin et al. found that the change in the angiogenic area of the filtering bleb at one month postoperatively could predict IOP at six months after surgery [35]. Hayek et al. found that conjunctival vascular density before surgery was much more strongly associated with postoperative IOP and surgical outcomes than the classical IBAGS [36].

To date, few studies have made full use of AS-OCTA to assess the vascular characteristic beneath the conjunctiva. We analyzed the vessels by depth to better evaluate the function of filtering blebs. Moreover, we compared the diagnostic efficiency between vessel parameters and traditional methods to estimate the application value of AS-OCTA.

## 2. Materials and Methods

### 2.1. Patient Selection

This was a cross-sectional study, conducted from December 2020 to June 2021 at Zhongshan Ophthalmic Center of Sun Yat-sen University. The study was authorized by the Institutional Review Board of Zhongshan Ophthalmic Center of Sun Yat-sen University (Ethic ID. 2020KYPJ119). The study was in accordance with the Declaration of Helsinki principles and was registered on Clinicaltrails.gov (NCT 04515017). All patients provided written informed consent.

For all patients enrolled, data were obtained on the age, sex, pre/postoperative IOP detected by Goldmann applanation tonometer (Haag-Streit, Bern, Switzerland), pre/postoperative topical ocular hypotensive drugs, and follow-up time. We only included one eye per patient. The inclusion and exclusion criteria were as follows: (1) age ≥ 18 years and ability to complete all examinations, (2) patients with primary open-angle glaucoma (POAG), and (3) trabeculectomy with MMC was performed over 6 months. Exclusion criteria: (1) ocular surgery other than trabeculectomy, (2) a history of ocular trauma, ocular surface inflammation, or other diseases that could influence the vascular status of the conjunctiva, Tenon’s capsule, and sclera layers, (3) the surgical area could not support high-quality imaging, and (4) suffering systemic disease, which made it impossible to complete the scanning for patients.

The criteria of surgical success were defined as a reduction in IOP ≥ 20% and postsurgery IOP ≤ 21 mmHg (with or without antiglaucoma medication).

We also classified the patients into three groups based on the usage of prostaglandin (PG). Groups A, B, and C included patients that did not use drugs, used drugs including PG, and used drugs without PG, respectively.

### 2.2. Surgical Technique

Glaucoma specialist (Y.H.Z.) performed all the limbus-based bleb trabeculectomy surgeries. Briefly, a 3 × 4 mm^2^ rectangular scleral flap was made and dissected to two-thirds of the scleral depth. MMC (0.4 mg/mL, SunRise, Shanghai, China) was applied for 2 min with balanced salt solution. Then, a piece of trabecular meshwork was excised, and peripheral iridectomy was performed. The scleral flap was sutured with two releasable sutures under mild tension, and the Tenon’s tissue was closed with 10-0 Ethilon (Ethicon, Johnson & Johnson Medical, San Angelo, TX, USA). Then, the conjunctiva was closed with a watertight suture using 8-0 Vicryl sutures (Ethicon, Johnson & Johnson Medical, San Angelo, TX, USA). If IOP was not well-controlled, the releasable sutures were removed. In addition, nd-doped yttrium aluminum garnet (ND-YAG) laser sutures lysis would be applied if the releasable sutures were eliminated.

### 2.3. Outcome Measures

#### 2.3.1. Bleb Evaluation Using OCTA

AS-OCTA Image Acquisition and Processing

AS-OCTA was performed by a single trained operator using OCTA scanning (6 × 6 mm^2^ HD Angio Disc protocol) with XR Avanti and AngioVue software (Optovue, Inc., Fremont, CA, USA). The Avanti OCT used the separated spectrum amplitude decorrelation algorithm (SSADA) to detect the blood flow in an acquired volume [37]. It used an 840 nm centered light source and was capable of 70,000 A-scans per second. The axial resolution of the device was 5 μm. The long corneal adapter (CAM-L) was needed for the anterior segment to obtain AS-OCTA images. For each bleb, a 6 × 6 mm^2^ acquisition was performed at the center of the scleral flap (Figure 1a,b). The size of the 6 × 6 mm^2^ scan pattern in the typical “HD Angio Disc” mode corresponded to 9 × 9 mm^2^ in AS-OCTA images. After adjusting the F and Z motor settings (with F = −15D, Z = +9.38) and canceling the automatic-tracking mode, the patient needed to look at the external fixed-eye position indicator throughout the whole scanning process.

En face images were delimited by two parallel curved lines separated manually. These two lines could be moved from the surface (conjunctival epithelium) to deeper layers (episclera) to explore the different layers of the filtering bleb. Superficial layer (SL), Tenon’s layer (TL), and deep layer (DL) vascular images could be acquired by setting the position from the conjunctival epithelium to a depth of 150 μm, from a depth of 150 μm to 250 μm, and from a depth of 150 μm to 1000 μm manually. The SL is mainly composed of the conjunctiva, and DL is composed of Tenon’s capsule and episclera.

The vessel data from AS-OCTA images were generated from the AS-OCTA instrument and quantified using ImageJ software (version 1.52a; http://imagej.nih.gov/ij, accessed on 16 January 2022). A signal strength index >60 was considered for the analysis [38]. Low-quality scans were defined as saccade or blinking artifacts disturbing vascularization analysis, and the captured part of the eyelid was excluded from the analysis. Three image files were required: SL, TL, and DL vascular images.

2.Quantitative Analysis

First, the binarization of images was acquired [34]. The binarization of superficial image was determined automatically by using Otsu threshold method in line with that previously reported [39]. However, this method detected excessive noise in the Tenon’s layer and deep-layer AS-OCTA images. Therefore, for the Tenon’s layer and deep layers, binarization was performed using Huang’s thresholding method (Figure 2).

Second, skeletonized images were acquired. A skeletonized vessel map was created by automatically skeletonizing the binarized images to transform vessels into a single-pixel line, regardless of its size or diameter (Figure 2). Both quantitative parameters were computed on the basis of these two kinds of images. Images were converted to the binary maps containing one and zero [40,41].

Vessel density (VD) was calculated as a unitless ratio of the total image area occupied by the vessels to the total image area after binarization of images, as follows:$$\mathrm{VD}=\frac{\sum_{i=1,\;j=1}^{n}A_{i,j}}{\sum_{i=1,j=1}^{n}X_{i,j}}$$

$A_{i,j}$ represented the number of pixels of vascularization (black pixels in Figure 2b,e,h), and $X_{i,j}$ represented all the pixels in the vascular region map (all pixels in Figure 2a,d,g). The pixel value of AS-OCTA image was expressed in (i, j) (assuming the size was 400 × 400 pixels).

Vessel diameter index (VDI) was calculated as ten times the ratio of the total image area, which was occupied by the skeletonized vessel area, defined as
$$\mathrm{VDI}=10\;\times \;\frac{\sum_{i=1,\;j=1}^{n}A_{i,j}}{\sum_{i=1,j=1}^{n}S_{i,j}}$$

$A_{i,j}$ indicated the number of pixels registered as vessel region (black pixels in Figure 2b,e,h) and $S_{i,j}$ indicated all the pixels registered in the skeletonized image (black pixels in Figure 2c,f,i). The pixel value of AS-OCTA image was expressed in (i, j) (assuming the size was 400 × 400 pixels) [40,41].

#### 2.3.2. Bleb Evaluation Using OCT

Bleb morphology was determined using corneal linear schemes of AS-OCT (AngioVue software; Optovue, Inc., Fremont, CA, USA). The scanning line was set to connect the midpoints of the two sides of the scleral flap, and the definition of bleb height was the longest distance between the conjunctival epithelium and scleral flap (Figure 3) [42].

The attending doctor (H.X.) examined the bleb vessel and morphological parameters (bleb height) in all patients. In addition, a resident doctor (M.L.) performed the image processing and analysis. All measurements were obtained from 2:30 p.m. to 5:00 p.m. to avoid the effects of IOP fluctuation. All values were performed three times, and the averages were used.

#### 2.3.3. Grading Using Conventional Bleb Grading Systems

Anterior segment photographs (SLE-8E; Chongqing Kang Hua Rui Ming Technology Co., Ltd., Shanghai, China) were taken with the patient gazing downward after the bleb location was confirmed by a review of the surgical record. Photographs were taken on the same day to compare the bleb evaluation method of OCTA with the conventional vascular grading systems (the KGS and IBAGS). The photographs were independently graded by a blinded attending doctor (Y.T.Z.). On the basis of KGS, the vessels of bleb were classified into the following categories: I, microvesicle type; II, diffuse flat type; III, scarring type; and IV, wrapper type. This study only utilized the vascularity scores from the IBAGS, which were classified as follows: V0, avascular white; V1, avascular cystic; V2, mild vascularity; V3, moderate vascularity; and V4, extensive vascularity.

### 2.4. Statistical Analysis

SPSS version 25.0 was used for analysis (SPSS Inc., Chicago, IL, USA). When calculating the sample size, the power was 90%, and the rate of type I error (a) was 5% (calculation methods are shown in Appendix A). The Shapiro–Wilk test was applied to test the distribution of variables. The normality data were represented as mean ± standard deviation (SD), and the quantitative measures of non-normality were described as median (interquartile range (IQR)) appropriately. The intergroup differences were analyzed by nonpaired *t*-test or one-way analysis of normally distributed variance. Kruskal–Wallis and rank-sum tests were used to evaluate the differences between non-normally distributed variables and categorical variables. Categorical variables were dummy-quantized (detailed in Appendix A). Pearson correlation or Spearman correlation analysis were applied appropriately. Tenon’s VD (TVD) and Tenon’s VDI (TVDI) were selected by stepwise regression (Appendix A). The factors affecting IOP included age, TVD, and TVDI, and the evaluation of KGS or IBAGS was analyzed by univariate and multivariate linear regression. Piecewise regression of TVD was performed at a value of 15% in TVD, which was determined by the loess plot. We performed univariate and multivariate logistic regression to explore the factors affecting surgical outcomes, including age, TVD, deep-layer VD (DVD), TVDI, deep-layer VDI (DVDI), and evaluation of KGS or IBAGS. Multivariable regression analysis was performed when *p*-value < 0.1. Receiver operating characteristic curve (ROC) analysis, area under the curve (AUC), and 95% confidence interval (CI) were performed for TVD, DVD, TVDI, DVDI, KGS, and IBAGS. A *p*-value < 0.05 was considered statistically significant.

## 3. Results

### 3.1. Patient Characteristics

A total of 54 patients conformed to the research condition initially. Among these patients, five were excluded due to poor image quality of AS-OCTA, which was caused by eye movement (motion artifacts). The other three patients were excluded because of the poor fitting degree of the artificial layering line. Ultimately, we included 46 eyes of 46 glaucoma patients for analysis; detailed information is shown in Appendix A.

The follow-up time was longer in the success group (*p* = 0.016, Table 1). Other baseline parameters, including age, sex, preoperative IOP, pre/postoperative topical ocular hypotensive drugs, and pre/postoperative PG administration, were not significantly different between the success and failure groups (Figure 4).

### 3.2. VD and VDI in Bleb Area

The VD and VDI of the SL, TL, and DL in the failure group were higher than those in the success group (*p* = 0.000, Table 2). However, there was no difference between the two groups in terms of bleb height (1.33 ± 0.46 mm versus 1.29 ± 0.54 mm, *p* = 0.852).

The associations between the VD and VDI in the three layers are presented in Table 3. There was a statistically significant correlation between IOP and the VDI in the SL, TL, and DL in the success group (*p* = 0.013, 0.016, and 0.031, respectively). Meanwhile, IOP was statistically significant correlated with the TVD, DVD, and TVDI (*p* = 0.012, 0.009, 0.028) in the failure group. However, the bleb height had no correlation with IOP in either group (*p* = 0.404, 0.808).

### 3.3. Predictive Factors for Surgical Outcomes

#### 3.3.1. Setting the IOP as the Dependent Variable

As shown in Table 4, the multiple linear regression analysis adjusted for the TVD, TVDI, and KGS revealed a significant association between the TVD (>15%), TVDI, and IOP (β [95%CI] = 0.667 (0.178–1.155), 0.720 (0.024–1.415), *p* = 0.009, 0.043). However, the KGS had no significant association with IOP. When the analysis was adjusted for the TVD, TVDI, and IBAGS, the TVD (>15%), TVDI, IBAGS X2, and IBAGS X3 were the factors most significantly associated with IOP (β (95%CI) = 0.630 (0.125–0.908), 0.788 (0.197–1.379), 7.540 (3.353–11.725), 11.249 (3.549–18.948), *p* = 0.011, 0.016, 0.001, 0.005, respectively).

#### 3.3.2. Setting an IOP of ≤21 mmHg as the Criteria for Surgical Success

Multiple logistic regression analysis revealed that the TVD and TVDI were correlated with the surgical outcomes (OR [95%CI] = 1.470 [1.037–2.085], 2.295 [1.009–5.224], *p* = 0.031, 0.048) (Table 5). There were no other covariates in the multiple analysis that had a *p* value <0.05.

### 3.4. Diagnostic Accuracy and Cut-Off Value

The ROC was drawn using the TVD, DVD, TVDI, DVDI, IBAGS, and KGS (Figure 5), and the diagnostic accuracy was measured by the AUC. The AUC (95% CI) were 0.960 (0.909–1.000), 0.948 (0.881–1.000), 0.925 (0.817–1.000), 0.911 (0.795–1.000), 0.770 (0.630–0.900), 0.850 (0.740–0.960), respectively.

The cut-off value of TVD, DVD, TVDI, DVDI, IBAGS, and KGS followed by the specificity and sensitivity were 21.002 (92.9%, 94.4%), 24.775 (89.3%, 94.4%), 13.577 (92.9%, 88.9%),12.562 (92.9%, 94.4%), 0.305 (64.3%, 77.8%), and 0.455 (89.3%, 72.2%) (Figure 5).

### 3.5. Influence of PG

The DVD was significantly different among the three groups (F = 3.456, *p* = 0.041). After Bonferroni adjustment, VD was higher in group A than in group B (*p* = 0.034). The comparison of SVD between groups A and C and between groups B and C showed no statistical difference (*p* = 1.000, 1.000). Additionally, SVD and TVD showed no statistical differences. The difference of VDI in the SL, TL, and DL among the three groups showed no statistical significance (Table 6).

## 4. Discussion

AS-OCTA provides a rapid, noninvasive, and quantitative technique for detecting vessel status in the internal site of the bleb. Our data indicate that the TVD and TVDI of filtering blebs detected by AS-OCTA could reflect the operation outcomes. The enhanced density and diameter of vessels beneath the conjunctiva were associated with a risk of failure. In addition, the AS-OCTA findings show better sufficient diagnostic accuracy than those obtained with the traditional clinical bleb analysis system. This new diagnostic technique could be promising for assessing filtering bleb function.

First, our study revealed that the failed cases presented denser and more dilated vessels in the bleb area than functional blebs. We found that the VDI was positively related with IOP if IOP ≤ 21 mmHg. In addition, VD showed a positive relation with increasing IOP if IOP went over 21 mmHg. This reminded us the VDI and VD might be related to the postoperative results, and a further prospective study would be needed for validation. Hayek et al. observed that, at one and six months postsurgery, the whole-layer VD of the filtering bleb significantly correlated with the IOP at the same follow-up [36]. Besides, previous studies found that a large proportion of successful blebs displayed high or moderate-height blebs [24,43,44]. Interestingly, in our study, the bleb height was similar between the success and failure groups, and the relationship between bleb height and IOP was not significant.

Moreover, our results indicate that the VD and VDI varied with depth. In particular, TVD and TVDI were most strongly correlated with the IOP and surgical outcome, which demonstrated the vascular biomarkers in TL might provide a more stable and effective assessment of filter bleb function. Failure of trabeculectomy was mainly caused by scar formation in the bleb site under the conjunctiva. In fact, angiogenesis was the main mechanism of scar formation after filtering surgery, and the enhanced density and diameter of vessels within scars was associated with surgical failure [12,14,27]. So, we speculated that the vascular parameters in the Tenon’s layer could better reflect the function of the filtering bleb. This was the advantage of AS-OCTA in observing vascular parameters under conjunctiva, which could not be detected by SLM, UBM, or AS-OCT.

In addition, we found that the increased TVD and TVDI were correlated with higher IOP and surgical failure, rather than bleb height, IBAGS, and KGS. IOP is the most frequently used index to evaluate surgery outcomes [45,46,47]. In line with the previous studies using in vivo confocal microscopy, our finding indicates that an avascular filtering bleb was related to the functional bleb. We predicted that IOP remained stable and low until VD reached 15%, and an increase in TVD and TVDI led to nearly double the risk of surgical failure. Hayek et al. observed that the whole-layer preoperative VD of the bleb was a good indicator of the operation success, and these parameters might be correlated with scar formation on Tenon’s capsules and the scleral flap [36]. A previous study showed that the VD at one month postoperatively was correlated with IOP six months postoperatively and bleb evaluation using OCTA demonstrated the possibility of using VD parameters to predict IOP [35]. In addition, VD at one week preoperation for filtering bleb was a good predictor of the reoperation [48]. Meanwhile, our study focused on the relationship between vascular parameters and surgical outcomes in the long-term follow-up. Interestingly, in our study, the cut-off value of DVD was similar to the preoperative density of patients with POAG [34,48]. Wound healing was required for two to three months after trabeculectomy as reported in [49,50]. According to previous studies, the VD increased in the first few months after surgery and then recovered to the presurgery level after three months [35,48]. These results are in line with our cut-off value, which supported that those vessel parameters detected by AS-OCTA (i.e., VD, VDI) were promising in predicting the function of the bleb.

The diagnostic efficiency of AS-OCTA vascular parameters was better than that of the traditional bleb classification and grading systems. The KGS and IBAGS had been widely used to evaluate the surface vessel visibility of filtering blebs by slit lamp [15,51]. We found that the KGS was not associated with the IOP or surgical results. Considering the IBAGS system, a higher grade of filtering bleb vessel showed a greater risk of IOP elevation, but it was not a significant factor affecting the surgical results. The VD and VDI in the TL and DL were sufficient to predict treatment results, because the sensitivity and specificity of the VD and VDI in determining the state of bleb were near 90%, and the AUC values were outstanding. In contrast, the KGS and IBAGS were subjective, qualitative, and poorly reproducible, and their AUC was relatively unsatisfactory. In previous studies, Hayek et al. showed a better correlation between VD and postoperative IOP results than that obtained with the IBAGS [36]. Nonetheless, Seo et al. found a good correlation between the VD of the bleb and the IBAGS vascularity grading, but neither was related to IOP [52]. Hence, the diagnostic efficiency might be improved by combining AS-OCTA and conventional evaluating methods of the filtering bleb.

In our study, PG administration led to an enhancement of VD in the DL. The side effects of PG administration must be considered. Akagi et al. observed that PG administration influenced the SVD significantly [53,54]. However, most of the patients were being treated with a number of different medications, so DVD might have been affected by their combined action. The major side reaction to PG administration was conjunctival hyperemia [55,56]. Preoperative hyperemia of the conjunctiva improves fibrosis due to the release of proinflammatory factors from these vessels [5,50,57]. Differences in the depth and time of PG on vasculature could be quantitively detected by AS-OCTA [54]. In addition, appropriate adjustment of eye drops based on AS-OCTA’s vessel parameters preoperatively might improve the success rate of surgery.

This study has several limitations. First, a prospective cohort study with a larger sample size should be conducted even though the sample size of the current study was sufficient based on scientific calculations. Second, at present, there is no specialized machine or software for AS-OCTA imaging and processing. This study and previous studies applied a posterior-pole module assisted by an external lens for anterior segment angiography and a general image processing program for vascular analysis. As artificial intelligence technology develops, more accurate image processing and analysis algorithms could help achieve a more comprehensive analysis of the vascular parameters of filtering blebs.

## 5. Conclusions

In conclusion, AS-OCTA provides a noninvasive quantitative evaluation of bleb vasculature, which has a significant impact on bleb function. A significant correlation between the vessel parameters in the TL and operative results suggests that AS-OCTA could be a promising tool for assessing the risk of failure in filtering surgery. In addition, compared with the traditional filtering bleb analysis systems, the VD and VDI showed superior diagnostic capability in assessing bleb function. Therefore, it appears that a quantitatively angiographic evaluation model detected by AS-OCTA would help clinicians analyze bleb function accurately and provide timely postoperative management.

## Figures and Tables

**Figure 1 jcm-11-01661-f001:**
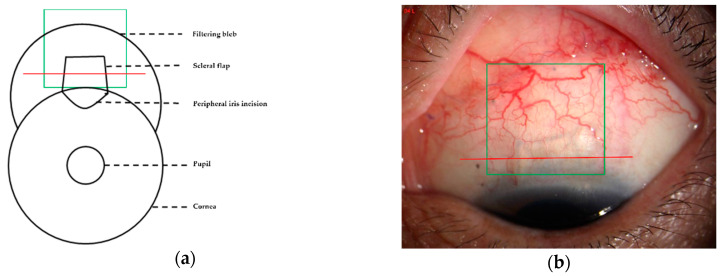
Acquisition of anterior segment optical coherence tomography (AS-OCT) and AS-OCT angiography (AS-OCTA). (**a**,**b**) HD Angio Disc scanning for 6 × 6 mm^2^ area performed in the center of the scleral flap (green boxes), and corneal linear schemes of AS-OCT (red lines).

**Figure 2 jcm-11-01661-f002:**
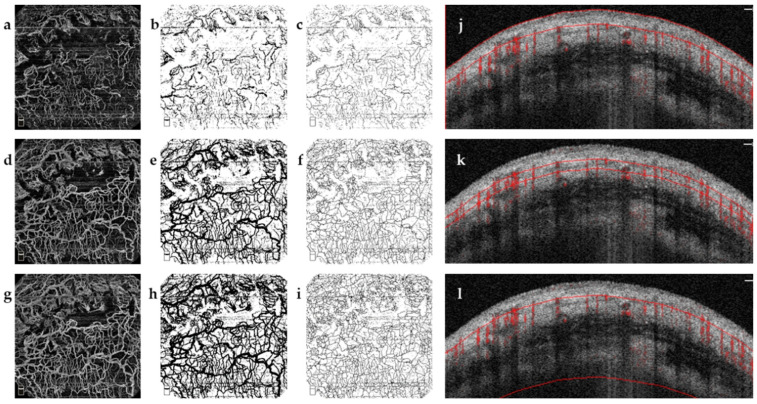
Image processing of anterior segment optical coherence tomography angiography (AS-OCTA). A functional filtering bleb image of right eye of a 63-year-old male patient at ten-year follow-up time point. Image processing was performed by ImageJ. (**a**,**d**,**g**) Vessel images of superficial layer (SL), Tenon’s layer (TL), and deep layer (DL) generated from the AS-OCTA. (**b**,**e**,**h**) Binarized images of SL, TL, and DL. Binarizing the angiography images into black and white pixels images for vessel density analysis. (**c**,**f**,**i**) Skeletonized images of SL, TL, and DL. Skeletonizing the binarized images to transform vessels into a single-pixel line for vessel diameter index analysis. (**a**–**c**,**j**) B-scan images of SL; red lines were determined as the depth of 0 and 150 microns from the conjunctival epithelium. (**d**–**f**,**k**) B-scan images of TL; red lines were determined as the depth of 150 and 250 microns from the conjunctival epithelium. (**g**–**i**,**l**) B-scan images of DL; red lines were determined as the depth of 150 and 1000 microns from the conjunctival epithelium.

**Figure 3 jcm-11-01661-f003:**
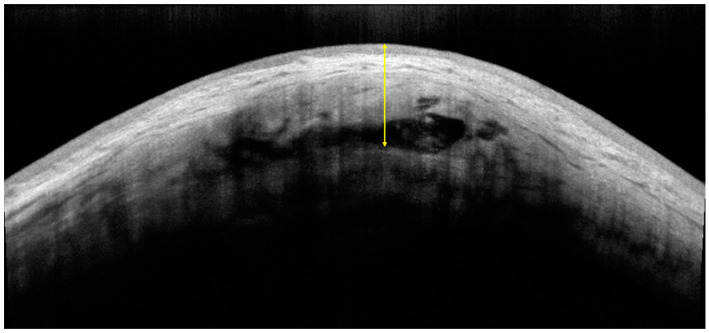
Bleb height evaluation method. Anterior segment optical coherence tomography (AS-OCT) image showing the measuring method of bleb height (yellow arrow lines).

**Figure 4 jcm-11-01661-f004:**
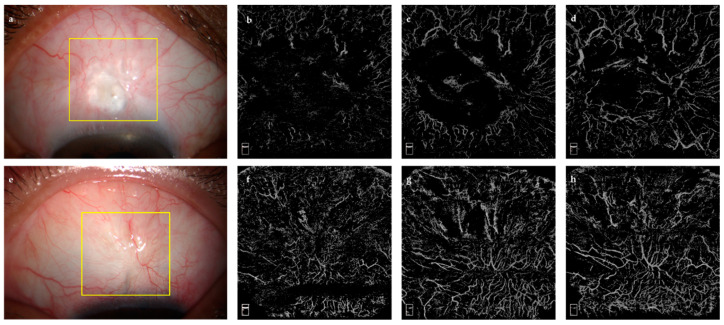
Cases showing anterior segment photographs and AS-OCTA images. (**a**) anterior segment photograph of success group; (**b**–**d**) superficial, Tenon’s, and deep-layer vessel flow image of success group; (**e**) anterior segment photograph of failure group; (**f**–**h**) superficial, Tenon’s, and deep-layer vessel flow image of failure group.

**Figure 5 jcm-11-01661-f005:**
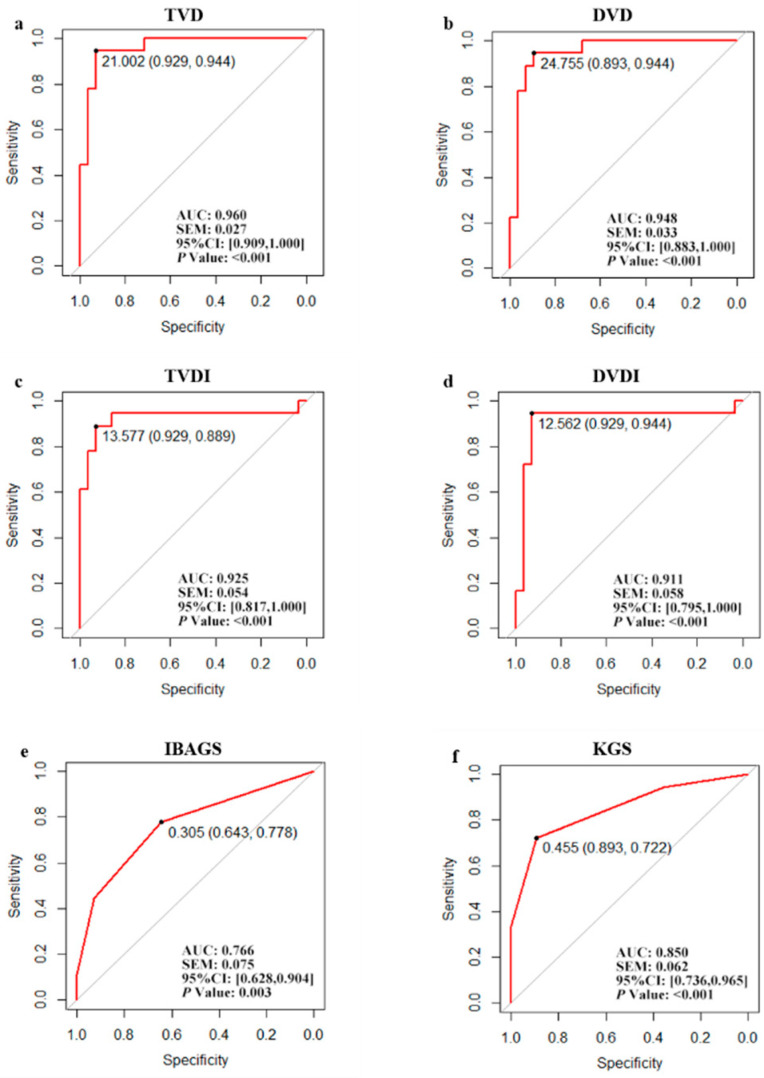
ROC for vessel parameters acquired using anterior segment optical coherence tomography angiography, IBAGS, and Kenfeld grading system. (**a**–**d**) ROC for TVD, DVD, TVDI, and DVDI. (**e**) ROC for IBAGS; (**f**) ROC for KGS. ROC = Receiver operating characteristic curve. AUC = area under curve. TVD = Tenon’s vessel density. TVDI = Tenon’s vessel diameter index. DVD = deep-vessel density. DVDI = deep-vessel diameter index. IBAGS = Indiana Bleb Grading Appearance Scale. KGS = Kenfeld grading system.

**Table 1 jcm-11-01661-t001:** Demographic and baseline characteristics of study subjects.

Variables	Success Group	Failure Group	*p* Value
	(28 Eyes)	(18 Eyes)	
Age (year)	48.50 [33.00]	48.50 [5.50]	0.973
Sex (male/female)	12/16	10/8	0.916
Preoperative IOP (mmHg)	25.58 (9.26)	24.92 (11.67)	0.857
Preoperative topical ocular hypotensive drugs (n)	1.00 [3.00]	1.00 [3.00]	0.215
Preoperative PG (n, %)	13.00 (46.43%)	6.00 (33.33%)	0.541
Follow-up time (year)	2.00 [9.00]	1.00 [2.25]	**0.008**
Postoperative topical ocular hypotensive drugs (n)	0.00 [1.00]	0.50 [3.00]	0.190
Postoperative PG (n, %)	7 (25.00%)	6 (33.33%)	0.540

IOP = intraocular pressure, PG = prostaglandin. Preoperative IOP is shown as the mean (standard deviation); the rest are shown as median (interquartile range) unless otherwise indicated. *p* values shown in bold are statistically significant. Comparing preoperative IOP between success and failure group using the unpaired *t*-test. Comparing age, pre/postoperative topical ocular hypotensive drugs using the rank-sum test. Comparing sex, pre/postoperative PG using the chi-square test.

**Table 2 jcm-11-01661-t002:** Vasculature parameters in bleb area between success and failure group.

Variables	Success Group (28 Eyes)	Failure Group (18 Eyes)	*p* Value
SL			
VD (%) ^1^	4.93 (3.36)	19.90 (5.82)	**<0.001**
VDI (pixel^−1^) ^2^	7.94 [2.08]	15.40 [1.48]	**<0.001**
TL			
VD (%) ^1^	10.94 (6.13)	27.83 (6.24)	**<0.001**
VDI (pixel^−1^) ^2^	8.65 [3.00]	16.97 [1.66]	**<0.001**
DL			
VD (%) ^1^	14.37 (7.70)	31.40 (6.25)	**<0.001**
VDI (pixel^−1^) ^2^	9.52 [3.29]	17.39 [2.16]	**<0.001**

SL = superficial layer; TL = Tenon’s layer; DL = deep layer; VD = vessel density; VDI = vessel diameter index. VD is shown as mean (standard deviation); VDI is shown as median (interquartile range). *p* values shown in bold are statistically significant. ^1^ Comparing VD between success and failure group using the unpaired *t*-test. ^2^ Comparing VDI between success and failure group using the rank-sum test.

**Table 3 jcm-11-01661-t003:** Correlation between the vascular parameters and IOP.

Factors	Success Group (28 Eyes, 28 Subjects)		Failure Group (18 Eyes, 18 Subjects)	
	r	*p* Value	r	*p* Value
SL				
VD (%) ^1^	0.016	0.936	−0.089	0.726
VDI (pixel^−1^) ^2^	0.462	**0.013**	0.273	0.274
TL				
VD (%) ^1^	−0.052	0.791	0.580	**0.012**
VDI (pixel^−1^) ^2^	0.450	**0.016**	0.517	**0.028**
DL				
VD (%) ^1^	0.092	0.642	0.597	**0.009**
VDI (pixel^−1^) ^2^	0.408	**0.031**	0.329	0.183

SL = superficial layer; TL = Tenon’s layer; DL = deep layer; VD = vessel density; VDI = vessel diameter index; IOP = intraocular pressure. *p* values shown in bold are statistically significant. ^1^ Pearson correlations between VD and IOP. ^2^ Spearman correlations between VDI and IOP.

**Table 4 jcm-11-01661-t004:** Univariate and multivariate linear regression analysis for IOP.

Variable	Univariable Model		Multivariable Model 1 ^3^		Multivariable Model 2 ^4^	
	Coefficient (95% CI)	*p* Value	Coefficient (95% CI)	*p* Value	Coefficient (95% CI)	*p* Value
Age (y)	0.083 (−0.107–0.268)	0.392	0.070 (−0.027–0.167)	0.151	0.057 (−0.060–0.173)	0.331
TVD ^1^						
<15%	0.332 (−0.319–0.982)	0.310	0.150 (−0.415–0.715)	0.595	0.114 (−0.593–0.820)	0.746
≥15%	**1.115 (0.768–1.462)**	**<0.001**	**0.630 (0.125–0.908)**	**0.011**	**0.667 (0.178–1.155)**	**0.009**
TVDI	**1.659 (1.250–2.069)**	**<0.001**	**0.788 (0.197–1.379)**	**0.016**	**0.720 (0.024–1.415)**	**0.043**
Bleb height	−0.745 (−5.853–4.386)	0.771	1.084 (−1.861–4.028)	0.460	2.503 (−1.586–6.591)	0.223
IBAGS ^2^						
2	**6.348 (1.858–10.838)**	**0.007**	1.266 (−2.279–4.811)	0.474		
3	**13.403 (7.981–18.826)**	**<0.001**	**7.540 (3.353–11.725)**	**0.001**		
4	**23.691 (13.990–33.391)**	**<0.001**	**3.797 (3.549–18.948)**	**0.005**		
KGS ^2^						
2	−0.196 (−5.539–5.148)	0.941			0.864 (−3.597–5.325)	0.223
3	**12.342 (6.832–17.851)**	**<0.001**			4.074 (−1.893–10.041)	0.806
4	**11.865 (5.261–18.469)**	**0.001**			1.990 (−4.301–8.282)	0.698

IOP = intraocular pressure; CI = confidence interval; TVD = Tenon’s vessel density; TVDI = Tenon’s vessel diameter index; IBAGS = Indiana Bleb Grading Appearance Scale; KGS = Kenfeld grading system. *p* values shown in bold are statistically significant. All variables with *p* < 0.05 in a univariable regression analysis were selected for multivariable regression analysis. ^1^ Piecewise regression of TVD is given at the value of 15% in TVD. ^2^ Categorical variables are dummy-quantized. ^3^ Adjusted for age, TVD, TVDI, bleb height, and IBAGS; ^4^ Adjusted for age, TVD, TVDI, bleb height, and Kenfeld.

**Table 5 jcm-11-01661-t005:** Univariate and multivariate logistic regression analysis for surgical outcome.

Variable	Univariable Model		Multivariable Model 1 ^3^		Multivariable Model 2 ^4^	
	OR (95% CI)	*p* Value	OR (95% CI)	*p* Value	OR (95% CI)	*p* Value
Age (y)	1.000 (0.961–1.043)	0.946				
TVD	**1.448 (1.165–1.799)**	**0.001**	**1.470 (1.037–2.085)**	**0.031**	**1.470 (1.037–2.085)**	**0.031**
TVDI	**1.862 (1.370–2.530)**	**<0.001**	**2.295 (1.008–5.224)**	**0.048**	**2.295 (1.008–5.224)**	**0.048**
Bleb height	0.895 (0.289–2.774)	0.848				
IBAGS ^1^						
2	3.37 (0.76–16.62)	0.115	0.012 (0.000–6.533)	0.169		
3	**13.5 (2.23–120.78)**	**0.008**	20.485 (0.258–1626.1)	0.176		
4	n/a ^2^	0.999	n/a ^2^	n/a ^2^		
KGS ^1^						
2	0.37 (0.02–3.02)	0.410			0.864 (−3.597–5.325)	0.387
3	**8.75 (1.67–58.61)**	**0.015**			4.074 (−1.893–10.041)	0.257
4	n/a ^2^	0.994			n/a ^2^	n/a ^2^

IOP = intraocular pressure; OR = odds ratio; CI = confidence interval; TVD = Tenon’s vessel density; TVDI = Tenon’s vessel diameter index; n/a = not applicable; IBAGS = Indiana Bleb Grading Appearance Scale; KGS = Kenfeld grading system. *p* values shown in bold are statistically significant. All variables with *p* < 0.05 in a univariable regression analysis were selected for multivariable regression analysis. ^1^ Categorical variables are dummy-quantized. ^2^ No sample was included in success group. ^3^ Adjusted for age, TVD, TVDI, and IBAGS; ^4^ Adjusted for age, TVD, TVDI, and KGS.

**Table 6 jcm-11-01661-t006:** The influence of prostaglandin administration on vasculature parameters.

Variables	Group A (29 Eyes, 29 Subjects)	Group B (13 Eyes, 13 Subjects)	Group C (4 Eyes, 4 Subjects)	*p* Value
VD (%) ^1^				
SL	9.27 (7.94)	14.11 (9.45)	11.00 (9.84)	0.248
TL	15.37 (10.00)	22.09 (9.44)	18.57 (13.08)	0.146
DL	18.13 (10.44)	27.29 (9.06)	21.76 (14.74)	**0.041**
VDI ^2^				
SL	9.54 [6.93]	11.32 [6.46]	9.55 [4.30]	0.264
TL	10.60 [8.06]	15.36 [6.11]	9.95 [5.54]	0.191
DL	11.59 [8.08]	16.61 [7.83]	10.52 [6.15]	0.206

VD = vessel density; VDI = vessel diameter index; SL = superficial layer; TL = Tenon’s layer; DL = deep layer; VD is shown as mean (standard deviation); VDI is shown as median (interquartile range). *p* values shown in bold are statistically significant. ^1^ Comparison using one-way ANOVA among group A, group B, and group C. ^2^ Comparison using Kruskal–Wallis test among group A, group B, and group C.

## Data Availability

The datasets generated and analyzed during the current study are available from the corresponding author on reasonable request.

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
