# Peer review of "Characteristic Assessment of Angiographies at Different Depths with AS-OCTA: Implication for Functions of Post-Trabeculectomy Filtering Bleb"

_jcm, 2022, doi:10.3390/jcm11061661_

Round 1
Reviewer 1 Report
This is an interesting paper as it touches on the opportunity of detecting bleb failure by means of non-invasive OCT technology. It is also interesting to divide the OCT scans in different layers to be more selective. Nevertheless, the paper has also several flaws:
1) English should definitely be improved. Just as an example, please see:
Not clear: “finished trabeculectomy….” Line 88. Not clear: systemic diseases than meant… line 92-3; Please also rewrite the sentence: when IOP came to… line 94-96A piece of trabecular meshwork and An iridectomy was applied (please rewrite). compositions (line 138); follicles (line 300).
Furthermore, the discussion needs to be revised as it is difficult to read. In the discussion, some sentences do not seem to be entirely supported by the data or by the literature.
The whole sentence (line 300-303) would be better rewritten.
It is also not entirely clear on which analysis, the Authors claim that their results predict an earlier change in the diameter than in the density.
The sentence in lines 323 and 326 is also difficult to understand. The paper seems to show that the univariate analysis correlated only the Tenon’s layer vascularity to IOP and success, but here you refer to a paper that claims that the trabeculectomy failure is due to deeper scar formation (14) and you did not find any correlation in this area. Please review the sentence from 337 and 342: how can the Authors say that previous studies did not categorize the patients according to the surgical outcomes, given that Yin reported OCTA to be predictive of post-operative IOP (categorization of patients) and Seo reported that one week VD was predictive of the trabeculectomy outcome?The Authors claim that the sample size is sufficient based on “scientific calculations” but they do not provide any evidence on this in the statistical section of the manuscript.The future development of AI analysis cannot be considered a limitation of this paper.
Aside from these observations, there are some methodological points that should definitely be described in more detail.
1) This is an 18 months prospective study: a) at which time point was the images taken? This is relevant as in Tab.1 the mean IOP of the failure group is 28,19 and it is not clear how such high IOPs would be tolerated. And it is surprising that the median medication count is 0,5 (this means that a patient was taking half the dosage of the medication or one drop ever other day?). Also if the images were taken at different time points, this might have influenced the analysis of the vessels.
2) In the same table the median post-operative (also, what the post-operative duration means exactly: the time when OCT was done? the total follow-up time?...) duration in the success group is 2 years, but this seems longer than the study time.
3) Could the Authors explain on which basis they decided to consider conjunctiva within 150 microns, tenon's between 151 (please correct in the paper) and 250 and the subtenon's space from 251 to 1000 microns? In case the bleb was higher than 1000 microns the deep layer were missed or this was not the case in any subject?
4) fig. 2 should be described in more detail in the legend as it is the "core" of the whole paper. For instance skeletonization is presented in b or in c?
Author Response
Responses to Reviewer’s comments
We greatly appreciate the reviewer’s insights and valuable comments. The reviewer’s concerns have been addressed in the revised manuscript. The changes made in the manuscript are highlighted in blue. Please find our point-by-point responses to all comments below. Moreover, we have performed the editing of English language and style.
Responses to the comments from Reviewers 1
(1) English should definitely be improved. Just as an example, please see. a) Not clear: “finished trabeculectomy….” Line 88. b) Not clear: systemic diseases than meant… line 92-3; c) Please also rewrite the sentence: when IOP came to… line 94-96; d) A piece of trabecular meshwork and An iridectomy was applied (please rewrite). e) compositions (line 138); f) follicles (line 300). g) The whole sentence (line 300-303) would be better rewritten.
Response: We greatly appreciate the reviewer’s helpful advice. We have carefully scrutinized the manuscript, and made corresponding revisions. The details have been shown as follows.
- a) We have cleared the sentence of “finished trabeculectomy….” in Line 88, and revised it as “(3) trabeculectomy with MMC was performed over 6 months” in Line 89.
- b) We have cleared the sentence “systemic diseases than meant…” in Line 92-3, and revised it as “(4) suffering systemic disease, which made patients impossible to complete the scanning” in Line 94-94.
- c) We have rewritten the sentence “when IOP came to…” in Line 94-96, and the revision was “The criteria of surgical success was defined as the reduction of IOP was ≥ 20%, and post-surgery IOP ≤ 21mmHg (with or without anti-glaucoma medication)” in Line 95-96.
- d) We have rewritten the sentence “A piece of trabecular meshwork and an iridectomy was applied” in Line 105, and we have revised it as “Then a piece of trabecular meshwork was excised and peripheral iridectomy was performed” in Line 104-105.
- e) We have revised the use of “compositions” in Line 138, and the sentence was shown as “The SL mainly composed of the conjunctiva, and DL composed of Tenon’s capsule and episclera” in Line 136-137.
- f) We have revised the word of “follicles”, and change it for “blebs” in Line 300. The sentenced has been shown as, “First, our study revealed that the failed cases presented denser and more dilated vessels in the bleb area than functional blebs.” in Line 302-303.
- g) The whole sentence in line 300-303 has been rewritten as follows, “VDI was positively related with IOP if IOP ≤ 21 mmHg. In addition, VD showed a positive relation with IOP increasing if IOP went over 21mmHg. This reminded us VDI and VD might be related to the postoperative results, and a further prospective study would be needed for validation.” in Line 303-306.
(2) It is also not entirely clear on which analysis; the Authors claim that their results predict an earlier change in the diameter than in the density.
Response: We greatly appreciate the reviewer’s helpful advice. We found that VDI was positively related with IOP if IOP ≤ 21 mmHg. In addition, VD showed a positive relation with IOP increasing if IOP went over 21mmHg. This reminded us VDI and VD might be related to the postoperative results, and a further prospective study would be needed for validation. We have revised relative parts in Line 303-306.
(3) The sentence in lines 323 and 326 is also difficult to understand. The paper seems to show that the univariate analysis correlated only the Tenon’s layer vascularity to IOP and success, but here you refer to a paper that claims that the trabeculectomy failure is due to deeper scar formation (14) and you did not find any correlation in this area. Please review the sentence.
Response: We greatly appreciate the reviewer’s helpful advice. we have revised it as follow, “Failure of trabeculectomy was mainly caused by scar formation in the bleb site under conjunctiva. In fact, angiogenesis was the main mechanism of scar formation after filtering surgery, and the enhanced density and diameter of vessels within scars was associated with surgical failure. So, we speculated that the vascular parameters in the Tenon’s layer could better reflect the function of the filtering bleb.” Please see the revised manuscript, Lines 316-321.
(1. Kim, M.; Lee, C.; Payne, R.; Yue, B. Y.; Chang, J. H.; Ying, H., Angiogenesis in glaucoma filtration surgery and neovascular glaucoma: A review. Surv Ophthalmol 2015, 60 (6), 524-35.)
(4) From 337 and 342: how can the Authors say that previous studies did not categorize the post-operative IOP (categorization of patients) and Seo reported that one week VD was predictive of the trabeculectomy outcome?
Response: We greatly appreciate the reviewer’s helpful advice to guide our revision and improve the quality of the manuscript. The expression is more appropriate as follows, “Previous study showed that the VD at one month postoperatively was correlated with IOP six months postoperatively and bleb evaluation using OCTA demonstrated the possibility of using VD parameters to predict IOP [51]. In addition, one week of VD preoperatively of filtering bleb was a good predictor of the reoperation [52]. Meanwhile, our study focused on the relationship between vascular parameters and surgical outcomes in long-term follow-up.”. It has been revised in the Lines 331-337.
(5) The Authors claim that the sample size is sufficient based on “scientific calculations” but they do not provide any evidence on this in the statistical section of the manuscript
Response: We greatly appreciate the reviewer’s helpful advice to guide our revision and improve the quality of the manuscript. We have supplemented the sample size calculation method in “Supplementary File 1”, and we also attached it on page 7.
(6) The future development of AI analysis cannot be considered a limitation of this paper.
Response: We greatly appreciate the reviewer’s helpful advice. After consideration, we think it is more rationale to list AI as the further prospect following the second point of limitation, which has been showed as follows,
Line 372-377. “Second, at present, there is no specialized machine or software for AS-OCTA imaging and processing. This study and previous studies, applied a posterior pole module assisted with an external lens for anterior segment angiography and a general image processing program for vascular analysis. As artificial intelligence technology develops, more accurate image processing and analysis algorithms could help achieve a more comprehensive analysis of the vascular parameters of filtering blebs.”
(7) This is an 18 months prospective study: a) at which time point was the images taken? b) This is relevant as in Tab.1 the mean IOP of the failure group is 28,19 and it is not clear how such high IOPs would be tolerated. c) And it is surprising that the median medication count is 0,5 (this means that a patient was taking half the dosage of the medication or one drop ever other day?). d) Also, if the images were taken at different time points, this might have influenced the analysis of the vessels.
Response: Thank you for your relevant and insightful questions. We need to correct the experimental design to be cross-sectional study. a) We collected these patients with postoperative outpatient re-examination in period of December 2020 to June 2021. b) The patients who had such high IOPs in failure group usually ran out of medications or failed to see the doctor in time. c) The number of patient in failure group was even number, so the calculation of the median required to average the two values that lie in the middle of all values as (0+1)/2=0.5. d) The postoperative time of patients was more than 6 months, and the postoperative recovery of filtering bleb has reached a stable state, including vascular state. We unified the time of detection from 2:30 p.m. to 5:00 p.m. to control the fluctuation of AS-OCTA parameters in one day.
(8) In the same table the median post-operative (also, what the post-operative duration means exactly: the time when OCT was done? the total follow-up time?...) duration in the success group is 2 years, but this seems longer than the study time.
Response: Thank you for your relevant and insightful questions. We have corrected the study design to the cross-sectional study. And we use “follow-up time” to replace “post-operative duration” to make the expression more accurate. “Follow-up time” means the time period between the date of operation and the date of AS-OCTA detection.
(9) a) Could the Authors explain on which basis they decided to consider conjunctiva within 150 microns, tenon's between 151 (please correct in the paper) and 250 and the subtenon's space from 251 to 1000 microns? b) In case the bleb was higher than 1000 microns the deep layer were missed or this was not the case in any subject?
Response: Thank you for your relevant and insightful questions. In order to express the meaning of each layer in our study, we rename the conjunctival layer to “superficial layer”, and the scleral layer to “deep layer”.
- a) First, we separated the AS-OCTA image into “superficial layer” and “deep layer” which were similar to previous studies. 1, 2 The “superficial layer” was mainly composed of the conjunctiva, and the “deep layer” was composed of Tenon’s and episcleral compositions. During the AS-OCTA scanning, we found that the vascular information at 150-250 microns was more closely related to the surgical results and postoperative IOP, so we added this layer and named it as Tenon’s layer.
b) There were 28 enrolled patients with the bleb higher than 1000 microns. Based on the scanning of our patients, we found the vascular signal was mainly distributed in 0-1000 as shown in B-scan angiography image attached below (Red lines were are determined as the depth of 0 and 1000 microns from the conjunctival epithelium. In addition, the red areas in Figure were vascular signals.) There were few vascular signals under 1000 microns, because this section composed of fluid and few episcleral vessels signal due to electrocoagulation during the operation. Therefore, we thought it was sufficient to observe 0-1000 microns vascular parameters of patients.
Reference:
- Akagi T, Uji A, Huang AS, et al. Conjunctival and Intrascleral Vasculatures Assessed Using Anterior Segment Optical Coherence Tomography Angiography in Normal Eyes. American Journal of Ophthalmology. 2018; 196:1-9.
- Akagi T, Uji A, Okamoto Y, et al. Anterior Segment Optical Coherence Tomography Angiography Imaging of Conjunctiva and Intrasclera in Treated Primary Open-Angle Glaucoma. American Journal of Ophthalmology. 2019; 208:313-22.
(10) Fig. 2 should be described in more detail in the legend as it is the "core" of the whole paper. For instance, skeletonization is presented in b or in c?
Response: Thank you for your relevant and insightful questions, and we have revised the legend of Figure 2 in Line 152-164. And the details were as follow, “Figure 2. Image processing of anterior segment optical coherence tomography angiography (AS-OCTA). A functional filtering bleb image of a 63-year-old male patient, with ten years follow-up time point on right eye. Image processing was performed by ImageJ. (a) (d) (g) vessel images of superficial layer (SL), Tenon’s layer (TL), and deep layer (DL) generated from the AS-OCTA. (b) (e) (h) binarized images of SL, TL, and DL. Binarizing the angiography images into black and white pixels images for vessel density analysis. (c) (f) (i) skeletonized images of SL, TL, and DL. Skeletonizing the binarized images to transform vessels into a single pixel line for vessel diameter index analysis. (a) (b) (c) (j) B-scan images of SL, red lines were determined as the depth of 0 and 150 microns from the conjunctival epithelium. (d) (e) (f) (k) B-scan images of TL, red lines were determined as the depth of 150 and 250 microns from the conjunctival epithelium. (g) (h) (i) (l) B-scan images of DL, red lines were determined as the depth of 150 and 1000 microns from the conjunctival epithelium”.
We sincerely hope that this revised manuscript has adequately addressed all of your comments and suggestions.
We have used two sample size calculation methods, as follows.
Methods 1 and 2 were calculated by gpower sample size calculation software
The sample size was expanded according to the 20% loss of follow-up rate.
Method 1 and 2 please see attachment

Reviewer 2 Report
The authors have presented a very interesting paper on a novel topic. It has clinical applicability. However, I have some comments that must be addressed before deciding on the suitability of the manuscript.
I have not reviewed the whole paper since I have major methodological comments that may significantly impact the results and the following discussion. More specifically, pre-operative medication must be accounted for (in general, and PG in particular) and the definition of success is not sufficiently good enough (<=21 mmHg with or without medication). Changing this definition may significantly impact the results.
-------------
Moderate English revision is needed throughout the study. I left some comments regarding this.
Abstract
- add “CVDI, TVDI, SVDI” as abbreviations after their respective anatomical layer in the previous sentence
- readers will not understand what “TVD (>15%)” is. Consider re-writing or removing from the abstract.
- “The area under curve (AUC)” for …? Please add here what was the prediction. Failure? Success?
- This sentence needs English revision “AS-OCTA provides a quantitative angiographic assessment of filtering bleb, especially for deep layer.”
Introduction
- overall, despite minor English adjustments needed, it is very well organized. The only thing that needs improvement is when transitioning to the last paragraph. It would be ideal to have a sentence stating what the knowledge gap is… what was not done in prior publications and what is missing.
Line 31 – add “corresponding” before “visual field defect”
Lien 39 – perhaps change “relevant to” for “associated with”
Line 48 – add references to the systems here as well.
Line 53 – “have been” instead of “are”
“In these studies, the filtering blebs with highly vascularization was the indicator of surgical failure.” – re-write
Line 62 – “are realized” - consider changing
Methods
Prospective and cross-sectional should be mutually exclusive, because prospective is many times considered as the same as longitudinal. This was not a prospective study since patients were evaluated only once.
How were patients selected? All consecutive patients that complied with inclusion/exclusion criteria?
Only in the Results section is apparent that some patients had several years of follow-up (IQR success group 9.0). This should have been clearer in the methods. Again, how were patients selected? Did the hospital only perform 54 trabeculectomies during 9 years?
Please state that you only included one eye per patient.
Line 84 – there is no mention to preoperative topical ocular hypotensive medications, but afterward a subgroup analysis is done regarding the preop use of PGs. I understand afterward that care was only given to “postoperative PG” use but preoperative medication should be accounted for when performing this analysis.
“When IOP came to ≤ 21 mmHg (with or without anti-glaucoma medication), the out- 94 come was considered successful. According to this rule, the group of success and failure 95 were defined.” – this is not enough. Please refer to the WGA guidelines for reporting glaucoma trials (https://wga.one/wga/guidelines-on-design-reporting-glaucoma-trials/). Ideally, there should be a % reduction (example 20%), otherwise a patient with preop IOP of 19mmHg (example) will always be successful. Please revise and adjust the results accordingly.
Line 104 – “which was not performed in elderly patients 104 with thin conjunctival tissue” does this mean there were patients included with and without the use of MMC? If not, since one of the inclusion criteria was using MMC, then please remove this part of the sentence.
Line 124 refers to using “Retina” mode but then Fig 1b) refers to HD Angio Disc.
Line 135-137: Since the trab leads to a variable height of bleb with very heterogeneous tenon’s thickness within, how were the depths for manual segmentation decided? Were any references used? Also, since there is an overlap of tenon’s (150-250) and sclera’s (150-1000) the same vessels are included twice. Why this choice with the overlap?
Figure 2’s correct legend seems to be missing.
Lines 210-214: why chose these parameters for the multivariate model specifically? Because of the results in univariate models? If so, then it needs to be explained. Usually variables are allowed in if p<0.10 in univariate model or if clinically significant irrespective of p value (eg. age).
Lines 210-214: several variables are never abbreviated. What is DVD? And DVDI?
Line 212: why chose 15% for the "Pricewise regression of TVD"?
Line 217: why chose these variables for the ROC model?
Results
Table 1 should report pre-operative medication as well, and ideally postoperative maneuveurs (needling, suture lysis, removal of sutures) should be reported as well.
“The IOP in the success group was higher than that of failure group” – this sentence does not make sense since IOP was the sole criterion to define the groups.
Author Response
Please see the attachment.
Responses to Reviewer’s comments
We greatly appreciate the reviewer’s insights and valuable comments. The reviewer’s concerns have been addressed in the revised manuscript. The changes made in the manuscript are highlighted in blue. Please find our point-by-point responses to all comments below. Moreover, we have performed the editing of English language and style.
Responses to the comments from Reviewers 2
(1) Abstract
- add “CVDI, TVDI, SVDI” as abbreviations after their respective anatomical layer in the previous sentence
- readers will not understand what “TVD (>15%)” is. Consider re-writing or removing from the abstract.
- “The area under curve (AUC)” for …? Please add here what was the prediction. Failure? Success?
- This sentence needs English revision “AS-OCTA provides a quantitative angiographic assessment of filtering bleb, especially for deep layer.”
Response: We greatly appreciate the reviewer’s helpful advice. We have carefully scrutinized the manuscript, and made corresponding revisions.
- Line 18-19. We have revised the “CVDI, TVDI, SVDI” as follows, “Significant correlations are found between SL, TL, DL’s VDI and IOP in the success group (p = 0.013, 0.016, 0.031), respectively.”
- Line 20. We have removed “(>15%)”, and the revision has been shown as follows, “Tenon’s VD (TVD) and Tenon’s VDI (TVDI) correlated with IOP adjusting for TVD, TVDI, and Indiana bleb appearance grading scale (IBAGS) (p = 0.009, 0.043) or Kenfeld grading system (KGS) (p = 0.011, 0.016).”
- Line 22-23. We have revised the sentence “The area under curve (AUC)…”, and added the prediction in the sentence. The revision has been shown as “The area under curve (AUC) of TVD, TVDI, IBAGS and KGS to predict the failure of surgery are 0.960, 0.925, 0.770 and 0.850.”
- Line 23-24. We have rewritten the sentence, and the revision was “AS-OCTA realizes the quantitative evaluation of vessels, especially the invisible vascularity beneath conjunctiva.”
(2) Introduction
- overall, despite minor English adjustments needed, it is very well organized. The only thing that needs improvement is when transitioning to the last paragraph. It would be ideal to have a sentence stating what the knowledge gap is… what was not done in prior publications and what is missing.
Response: Thank you for your relevant and insightful questions. We have added the knowledge gap as follows, “To date, few researches have made the full use of AS-OCTA to assess the vascular characteristic beneath conjunctiva. We analyzed the vessels by depth to better evaluate the function of filtering blebs. Moreover, we compared the diagnostic efficiency between vessel parameters and traditional methods to estimate the application value of AS-OCTA.” in Line 71-75.
(3) Introduction
Line 31 – add “corresponding” before “visual field defect”
Lien 39 – perhaps change “relevant to” for “associated with”
Line 48 – add references to the systems here as well.
Line 53 – “have been” instead of “are”
“In these studies, the filtering blebs with highly vascularization was the indicator of surgical failure.” – re-write
Line 62 – “are realized” - consider changing
Response: We greatly appreciate the reviewer’s helpful advice. We have carefully scrutinized the manuscript, and made corresponding revisions.
Line 31 – We have added “corresponding” before “visual field defect”, and the revision is “the progressive damage of the optic nerve and corresponding visual field defect is the characteristic changes of glaucoma” in Line 32.
Line 39 – We have changed “relevant to” for “associated with”, and the revision is “Maintaining the function of postoperative filtering blebs is associated with the degree of conjunctival fibrosis, tissue remodeling, and angiogenesis in the surgical area” in Line 40.
Line 48 – We have added references [15-18] to the systems in the Line 50.
Line 53 – We have changed “are” for “have been”, and the revision is “anterior segment optical coherence tomography (AS-OCT) has been applied to show the internal structure of bleb” in Line 54.
Line 53 – We have re-written the sentence as follows, “In these studies, highly vascularized bleb was an indicator of surgical failure.” In Line 59.
Line 62 – We have changed the “are realized” for “has been achieved”. The sentenced has been revised as “Objective and quantitative assessment of vascular features has been achieved by optical coherence tomography angiography (OCTA) noninvasively” in Line 63.
(4) Methods
Prospective and cross-sectional should be mutually exclusive, because prospective is many times considered as the same as longitudinal. This was not a prospective study since patients were evaluated only once.
Response: Thank you for your relevant and insightful questions. We need to correct the clinical design to be cross-sectional study. Please see the revised manuscript, Lines 78.
(5) Methods
How were patients selected? All consecutive patients that complied with inclusion/exclusion criteria?
Only in the Results section is apparent that some patients had several years of follow-up (IQR success group 9.0). This should have been clearer in the methods. Again, how were patients selected? Did the hospital only perform 54 trabeculectomies during 9 years?
Please state that you only included one eye per patient.
Response: Thank you for your relevant and insightful questions. Our previous statement was not clear enough, and our study was a cross-sectional study.
Our study time was between December 2020 and June 2021, and we chose the same specialist clinic to enroll patients. Following our inclusion and exclusion criteria, we enrolled 54 patients who had already been performed trabeculectomy over 6 months, but the follow-up time point was different. Follow-up time for every patient was supplemented in Supplementary Table S2, and also has been attached on page 10-11.
Among these patients, five were excluded due to poor image quality of AS-OCTA, which was caused by eye movement (motion artifacts). The other three patients were excluded because of the poor fitting degree of the artificial layering line. Ultimately, we included 46 eyes of 46 glaucoma patients for analysis.
We have added the state that “we only included one eye per patient” in Line 86-87.
(6) Line 84 – there is no mention to preoperative topical ocular hypotensive medications, but afterward a subgroup analysis is done regarding the preop use of PGs. I understand afterward that care was only given to “postoperative PG” use but preoperative medication should be accounted for when performing this analysis.
Response: We thank the reviewer for the helpful advice and comprehensive review to guide our revision. We supplemented the preoperative medications, and the preoperative use of PG in Table 1. The details of each patient have been added in Supplementary Table S2, and also has been attached on page 10-11.
(7) “When IOP came to ≤ 21 mmHg (with or without anti-glaucoma medication), the out- 94 come was considered successful. According to this rule, the group of success and failure 95 were defined.” – this is not enough. Please refer to the WGA guidelines for reporting glaucoma trials (https://wga.one/wga/guidelines-on-design-reporting-glaucoma-trials/). Ideally, there should be a % reduction (example 20%), otherwise a patient with preop IOP of 19mmHg (example) will always be successful. Please revise and adjust the results accordingly.
Response: Thank you for your relevant and insightful questions. We have revised it as follows, “The criteria of surgical success was defined as the reduction of IOP was ≥ 20%, and post-surgery IOP ≤ 21mmHg (with or without anti-glaucoma medication).” in the Lines 95-96.
We found that patients who originally belonged to the success group still met the new definition, so the results did not change according to the updated criteria. Preoperative IOP has been added in Table 1, and the details of each patient has been added in Supplementary Table S2. We also has attached the details on page 10-11.
(8) Line 104 – “which was not performed in elderly patients 104 with thin conjunctival tissue” does this mean there were patients included with and without the use of MMC? If not, since one of the inclusion criteria was using MMC, then please remove this part of the sentence.
Response: We thank you for your relevant and insightful questions. We have reviewed “Surgical Technique”, and all patients were assisted with MMC during the operation. We have changed the description of the operation process in Line 102-103.
(9) Line 124 refers to using “Retina” mode but then Fig 1b) refers to HD Angio Disc.
Response: Thank you for your relevant and insightful questions. We have corrected the scanning mode in Line 123 as “HD Angio Disc”.
(10) a) Line 135-137: Since the trab leads to a variable height of bleb with very heterogeneous tenon’s thickness within, how were the depths for manual segmentation decided? Were any references used? b) Also, since there is an overlap of tenon’s (150-250) and sclera’s (150-1000) the same vessels are included twice. Why this choice with the overlap?
Response: Thank you for your relevant and insightful questions. In order to express the meaning of each layer in our study, we have renamed the conjunctival layer to superficial layer, and the scleral layer to deep layer.
- a) First, we separated the AS-OCTA image into superficial layer and deep layer as previous studies. 1, 2 The “superficial layer” was mainly composed of the conjunctiva, and the “deep layer” was composed of Tenon’s capsule and episclera.
Secondly, based on the scanning of our patients, we found the vascular information in the center of filtering bleb was mainly distributed in 0-1000 microns though the trabeculectomy could lead to a variable height of bleb, as shown in B-scan angiography image attached below (Red lines were are determined as the depth of 0 and 1000 microns from the conjunctival epithelium. In addition, the red areas in Figure were vascular signals).
In addition, our approach reduced subjective bias.
- b) During analysis, we noticed vessel density and diameter in Tenon’s layer (150-250 microns) was most associated with postoperative high intraocular pressure and surgical failure. So, we
separated this layer for analysis.
- Akagi T, Uji A, Huang AS, et al. Conjunctival and Intrascleral Vasculatures Assessed Using Anterior Segment Optical Coherence Tomography Angiography in Normal Eyes. American Journal of Ophthalmology. 2018; 196:1-9.
- Akagi T, Uji A, Okamoto Y, et al. Anterior Segment Optical Coherence Tomography Angiography Imaging of Conjunctiva and Intrasclera in Treated Primary Open-Angle Glaucoma. American Journal of Ophthalmology. 2019; 208:313-22.
(11) Figure 2’s correct legend seems to be missing.
Response: Thank you for your relevant and insightful questions, and we have revised the legend of Figure 2 in Line 152-164. And the details were as follow, “Figure 2. Image processing of anterior segment optical coherence tomography angiography (AS-OCTA). A functional filtering bleb image of a 63-year-old male patient, with ten years follow-up time point on right eye. Image processing was performed by ImageJ. (a) (d) (g) vessel images of superficial layer (SL), Tenon’s layer (TL), and deep layer (DL) generated from the AS-OCTA. (b) (e) (h) binarized images of SL, TL, and DL. Binarizing the angiography images into black and white pixels images for vessel density analysis. (c) (f) (i) skeletonized images of SL, TL, and DL. Skeletonizing the binarized images to transform vessels into a single pixel line for vessel diameter index analysis. (a) (b) (c) (j) B-scan images of SL, red lines were determined as the depth of 0 and 150 microns from the conjunctival epithelium. (d) (e) (f) (k) B-scan images of TL, red lines were determined as the depth of 150 and 250 microns from the conjunctival epithelium. (g) (h) (i) (l) B-scan images of DL, red lines were determined as the depth of 150 and 1000 microns from the conjunctival epithelium”.
(12) Lines 210-214: why chose these parameters for the multivariate model specifically? Because of the results in univariate models? If so, then it needs to be explained. Usually, variables are allowed in if p<0.10 in univariate model or if clinically significant irrespective of p value (eg. age).
Response: Thank you for your relevant and insightful questions, and we agree with reviewer’s comments. In univariate regression analysis, we included all vascular parameters. Then we selected vascular parameters to enter multivariate regression analysis with p-value < 0.1. For the same type of vascular parameters (e.g., SVDI, TVDI, and DVDI), we used forward stepwise regression to choose the most appropriate parameters entering the multivariate regression analysis. Stepwise regression required a entering factor with p-value < 0.05 then removing factors with p-value < 0.1. We have added relevant statistical tables in the Supplementary Table S3 and S4, and attached them on page 12 and page 13.
(13) Lines 210-214: several variables are never abbreviated. What is DVD? And DVDI?
Response: We thank you for your relevant and insightful questions. We have added the abbreviation in Line 226-227.
(14) Line 212: why chose 15% for the "Pricewise regression of TVD"?
Response: We thank you for your relevant and insightful questions. We used TVD as the independent variable and IOP as the dependent variable to draw the loess plot. The trend showed that when TVD was less than 15%, TVD remains stable and had nothing to do with the rise of IOP. When TVD was greater than or equal to 15%, TVD was positively correlated with IOP. So, we took TVD = 15% as the boundary for pricewise regression. The loess plot has been showed as follows,
(15) Line 217: why chose these variables for the ROC model?
Response: We thank the reviewer for the helpful advice and comprehensive review to guide our revision. After univariate logistic regression, it was found that CVD and CVDI were not associated with surgical outcomes, so they were not included in ROC model. TVD, TVDI, DVD, DVDI, IBAGS and KGS were statistically significant, and chosen for ROC model.
On the other hand, we want to calculate AUC through ROC curve, and compare the diagnostic efficiency of between AS-OCTA vascular parameters and conventional grading systems. We supplemented the results of univariate logistic regression in Supplementary Table S4, and we also have attached them on page 13.
(16) Results
Table 1 should report pre-operative medication as well, and ideally postoperative maneuveurs (needling, suture lysis, removal of sutures) should be reported as well.
Response: We thank the reviewer for the helpful advice and comprehensive review to guide our revision. We have supplemented the preoperative medication of patients in Supplementary Table S2, and also attached it on page 10-11.
None of the 46 patients included in the study received needling, and removal of sutures. Suture lysis was completed in the early stage within 3-4 weeks after operation. Our study included the patient who had been performed surgery over 6 months, and the filtering bleb had reached the stable state, which would be less affected by early postoperative maneuvers.
(17) Results
” The IOP in the success group was higher than that of failure group” – this sentence does not make sense since IOP was the sole criterion to define the groups.
Response: Thank you for your relevant and insightful questions. We also agree that the analysis of IOP between the two groups has limited practical significance, so we have deleted this part.
We sincerely hope that this revised manuscript has adequately addressed all of your comments and suggestions.
|
Supplementary Table S2. Detailed Baseline Characteristics of Study Subjects |
||||||||
|
Follow-up time (year) |
Preoperative IOP (mmHg) |
Postoperative IOP (mmHg) |
The percentage of IOP reduction (%) |
Preoperative topical ocular hypotensive drugs (n) |
Preoperative use of PGs (n) |
Postoperative topical ocular hypotensive drugs (n) |
Postoperative use of PGs (n) |
|
1 |
1.5 |
14 |
7 |
47 |
2 |
0 |
0 |
0 |
|
2 |
0.5 |
14 |
7 |
49 |
0 |
0 |
0 |
0 |
|
3 |
5.0 |
24 |
18 |
25 |
3 |
1 |
0 |
0 |
|
4 |
1.0 |
25 |
13 |
47 |
0 |
0 |
1 |
1 |
|
5 |
3.0 |
22 |
14 |
35 |
3 |
1 |
0 |
0 |
|
6 |
11.0 |
36 |
17 |
53 |
3 |
1 |
1 |
0 |
|
7 |
10.0 |
41 |
14 |
65 |
3 |
1 |
1 |
1 |
|
8 |
10.0 |
20 |
15 |
27 |
3 |
1 |
1 |
1 |
|
9 |
6.0 |
21 |
5 |
77 |
3 |
1 |
0 |
0 |
|
10 |
1.0 |
19 |
13 |
30 |
3 |
1 |
0 |
0 |
|
11 |
10.0 |
17 |
12 |
28 |
3 |
1 |
0 |
0 |
|
12 |
2.0 |
17 |
6 |
63 |
0 |
0 |
0 |
0 |
|
13 |
0.5 |
13 |
7 |
43 |
1 |
0 |
0 |
0 |
|
14 |
10.0 |
22 |
16 |
29 |
0 |
0 |
2 |
1 |
|
15 |
2.0 |
26 |
18 |
32 |
0 |
0 |
0 |
0 |
|
16 |
2.0 |
36 |
19 |
48 |
0 |
0 |
0 |
0 |
|
17 |
1.0 |
24 |
17 |
28 |
3 |
1 |
0 |
0 |
|
18 |
1.0 |
21 |
11 |
46 |
3 |
1 |
0 |
0 |
|
19 |
1.5 |
35 |
14 |
60 |
0 |
0 |
0 |
0 |
|
20 |
2.5 |
17 |
11 |
32 |
0 |
0 |
2 |
0 |
|
21 |
5.0 |
40 |
16 |
60 |
0 |
0 |
0 |
0 |
|
22 |
1.5 |
38 |
10 |
74 |
0 |
0 |
0 |
0 |
|
23 |
20.0 |
48 |
19 |
60 |
3 |
0 |
3 |
1 |
|
24 |
20.0 |
25 |
19 |
23 |
0 |
0 |
3 |
1 |
|
25 |
1.0 |
28 |
19 |
32 |
0 |
0 |
0 |
0 |
|
26 |
14.0 |
20 |
9 |
53 |
3 |
1 |
3 |
1 |
|
27 |
2.0 |
29 |
12 |
59 |
3 |
1 |
0 |
0 |
|
28 |
0.5 |
26 |
14 |
46 |
3 |
1 |
0 |
0 |
|
29 |
0.5 |
23 |
39 |
-70 |
0 |
0 |
1 |
0 |
|
30 |
1.0 |
32 |
38 |
-19 |
3 |
1 |
3 |
1 |
|
31 |
1.0 |
44 |
36 |
18 |
2 |
0 |
3 |
1 |
|
32 |
1.0 |
41 |
28 |
31 |
0 |
0 |
1 |
0 |
|
33 |
0.5 |
19 |
27 |
-42 |
3 |
1 |
1 |
0 |
|
34 |
0.5 |
16 |
24 |
-51 |
3 |
1 |
0 |
0 |
|
35 |
5.0 |
19 |
23 |
-21 |
0 |
0 |
0 |
0 |
|
36 |
1.0 |
45 |
43 |
5 |
1 |
0 |
0 |
0 |
|
37 |
0.5 |
40 |
25 |
37 |
3 |
0 |
0 |
0 |
|
38 |
12.0 |
19 |
28 |
-46 |
1 |
0 |
3 |
1 |
|
39 |
12.0 |
26 |
23 |
10 |
2 |
0 |
3 |
1 |
|
40 |
2.0 |
25 |
23 |
6 |
0 |
0 |
0 |
0 |
|
41 |
0.5 |
22 |
22 |
2 |
3 |
1 |
0 |
0 |
|
42 |
2.0 |
23 |
26 |
-13 |
0 |
0 |
0 |
0 |
|
43 |
2.0 |
24 |
30 |
-25 |
0 |
0 |
0 |
0 |
|
44 |
5.0 |
35 |
24 |
31 |
3 |
1 |
3 |
1 |
|
45 |
0.5 |
28 |
22 |
21 |
0 |
0 |
0 |
0 |
|
46 |
1.0 |
28 |
26 |
7 |
1 |
0 |
3 |
1 |
|
IOP = intraocular pressure, PG= prostaglandin.
|
|||||||||
Supplementary Table S3. Univariate and Multivariate Linear Regression Analysis for IOP |
||||||
Variable |
Univariable Model |
Multivariable Model 1* |
Multivariable Model 2+ |
|||
|
Coefficient (95% CI) |
p Value |
Coefficient (95% CI) |
p Value |
Coefficient (95% CI) |
p Value |
SL |
||||||
VD (%) |
0.515 (-0.336-1.365) |
0.229 |
||||
VDI (pixel-1) |
1.143 (-0.085-2.201) |
0.035 |
n/a |
0.15 |
||
TL |
||||||
VD (%) |
0.648 (0.472-0.823) |
0.000 |
0.648 (0.472-0.823) |
0.000 |
||
VDI (pixel-1) |
1.659 (1.250-2.069) |
<0.001 |
1.659 (1.250-2.069) |
0.000 |
||
DL |
||||||
VD (%) |
0.599 (0.431-0.767) |
0.000 |
n/a |
0.677 |
||
VDI (pixel-1) |
1.526 (1.109-1.943) |
0.000 |
n/a |
0.183 |
||
IOP= intraocular pressure; SL = superficial layer; TL = Tenon’s layer; DL = deep layer; CI= confidence interval; VD= Vessel density; VDI= Vessel diameter index; n/a= not applicable. P values are shown in bold as statistically significant. All variables with p < 0.1 in a univariable regression analysis was selected for multivariable regression analysis. * Stepwise regression for VD in TL and DL. + Stepwise regression for VDI in SL, TL, and DL. ﹟Not included in multivariate model after stepwise regression. |
Supplementary Table S4. Univariate and Multivariate Logistic Regression Analysis for Surgical Outcome |
||||||
Variable |
Univariable Model |
Multivariable Model 1* |
Multivariable Model 2+ |
|||
|
OR (95% CI) |
P Value |
OR (95% CI) |
P Value |
OR (95% CI) |
P Value |
Conjunctival layer |
|
|
|
|
|
|
VD (%) |
1.192 (0.972-1.460) |
0.091 |
n/a﹟ |
0.175 |
|
|
VDI (pixel-1) |
1.312 (0.970-1.774) |
0.078 |
|
|
n/a |
0.595 |
Tenon's layer |
|
|
|
|
|
|
VD (%) |
1.448 (1.165-1.799) |
0.001 |
1.448 (1.165-1.799) |
0.001 |
|
|
VDI (pixel-1) |
1.862 (1.370-2.530) |
0.000 |
|
|
1.862 (1.370-2.530) |
0.000 |
Scleral layer |
|
|
|
|
|
|
VD (%) |
1.308 (1.128-1.517) |
0.000 |
n/a﹟ |
0.376 |
|
|
VDI (pixel-1) |
1.724 (1.315-2.260) |
0.000 |
|
|
n/a |
0.105 |
OR= odds ratio; CI= confidence interval; VD= Vessel density; VDI= Vessel diameter index; n/a= not applicable; n/a= not applicable. P values are shown in bold as statistically significant. All variables with p < 0.1 in a univariable regression analysis was selected for multivariable regression analysis. * Stepwise regression for conjunctival, Tenon’s and scleral VD. + Stepwise regression for conjunctival, Tenon’s and scleral VDI. ﹟Not included in multivariate model after stepwise regression. |

Reviewer 3 Report
line 18, the words of the acronyms CVDI, TVDI, SVDI are not explained.
line 22: the words of the abbreviations IBAGS and KGS are not explained
In the discussion:
line 301 and following: 'IOP was lower 21...exceeded 21..': the argument is confusing and contradictory. It needs to be better explained.
line 317: 'TVDI was positively correlated...'. Repeat of line 310. More repetition on line 327. All speech from line 297 to 327 is verbose and repetitive, needs to be condensed and more clearly expressed
line 351 and 360; two repeated sentences which express the same concept
Author Response
Responses to Reviewer’s comments
We greatly appreciate the reviewer’s insights and valuable comments. The reviewer’s concerns have been addressed in the revised manuscript. The changes made in the manuscript are highlighted in blue. Please find our point-by-point responses to all comments below. Moreover, we have performed the editing of English language and style.
Responses to the comments from Reviewers 3
(1) line 18, the words of the acronyms CVDI, TVDI, SVDI are not explained.
line 22: the words of the abbreviations IBAGS and KGS are not explained
Response: We thank the reviewer for the helpful advice and comprehensive review to guide our revision. We have added explanation of the abbreviation of “CVDI, TVDI, SVDI” in Line 20-23.
(2) In the discussion:
line 301 and following: ‘IOP was lower 21…exceeded 21...': the argument is confusing and contradictory. It needs to be better explained.
Response: We greatly appreciate the reviewer’s helpful advice. We have revised relative parts as follows, “VDI was positively related with IOP if IOP ≤ 21 mmHg. In addition, VD showed a positive relation with IOP increasing if IOP went over 21mmHg. This reminded us VDI and VD might be related to the postoperative results, and a further prospective study would be needed for validation.” in Line 303-306.
(3) a) line 317: 'TVDI was positively correlated...'. Repeat of line 310. More repetition on line 327.
- b) All speech from line 297 to 327 is verbose and repetitive, needs to be condensed and more clearly expressed
- c) line 351 and 360; two repeated sentences which express the same concept
Response: Thank you for your relevant and insightful questions.
- a) We have revised the sentences in line 310, 317, and 327. In addition, we have re-written these sentences as follows.
Line 303-304: “VDI was positively related with IOP if IOP ≤ 21 mmHg”.
Line 313-316: “In particular, TVD and TVDI were most strongly correlated with the IOP and surgical outcome, which demonstrated the vascular biomarkers in TL might provide a more stable and effective assessment of filter bleb function.”.
Line 323-325: “In addition, we found that the increased TVD and TVDI were correlated with higher IOP and surgical failure, other than bleb height, IBAGS, and KGS. IOP is the most frequently used index to evaluated the surgery outcomes.”.
- b) We have revised all speech from line 297 to 327, and we have condensed the expression in Line 302-327.
- c) We have removed the sentence “The vessel parameters detected by AS-OCTA were more predictable than those of the traditional systems” in Line 360.
We sincerely hope that this revised manuscript has adequately addressed all of your comments and suggestions.